

# 1 Nitrate-driven haze pollution during summertime over the North China
# 2 Plain

Haiyan Li[1], Qiang Zhang[2], Bo Zheng[3], Chunrong Chen[2], Nana Wu[2], Hongyu Guo[4], Yuxuan Zhang[2], Yixuan
Zheng[2], Xin Li[2], Kebin He[1,5]
[1] State Key Joint Laboratory of Environment Simulation and Pollution Control, School of Environment, Tsinghua University,
Beijing 100084, China
[2] Ministry of Education Key Laboratory for Earth System Modeling, Department of Earth System Science, Tsinghua University,
Beijing 100084, China
[3] Laboratoire des Sciences du Climat et de l'Environnement, CEA-CNRS-UVSQ, UMR8212, Gif-sur-Yvette, France
[4] School of Earth and Atmospheric Sciences, Georgia Institute of Technology, Atlanta, GA, 30332, USA
[5] State Environmental Protection Key Laboratory of Sources and Control of Air Pollution Complex, Tsinghua University, Beijing
100084, China
*Correspondence to:* Qiang Zhang (qiangzhang@tsinghua.edu.cn) or Kebin He (hekb@tsinghua.edu.cn)
**Abstract**. Compared to the severe winter haze episodes in the North China Plain (NCP), haze pollution during summertime has
drawn little public attention. In this study, we present the highly time-resolved chemical composition of submicron particles ($PM_1$)
measured in Beijing and Xinxiang in the NCP region during summertime to evaluate the driving factors of aerosol pollution.
During the campaign periods (30 June to 27 July, 2015, for Beijing and 8 to 25 June, 2017, for Xinxiang), the average $PM_1$
concentrations were 35.0 μg m$^{-3}$ and 64.2 μg m$^{-3}$ in Beijing and Xinxiang, respectively. Pollution episodes characterized with
largely enhanced nitrate concentrations were observed at both sites. In contrast to the slightly decreased mass fractions of sulfate,
semi-volatile oxygenated organic aerosol (SV-OOA), and low-volatile oxygenated organic aerosol (LV-OOA) in $PM_1$, nitrate
displayed an almost linearly increased contribution with the aggravation of aerosol pollution in both Beijing and Xinxiang,
highlighting the importance of nitrate formation as the driving force of haze evolution in summer. Rapid nitrate production mainly
occurred after midnight, with a higher formation rate than that of sulfate, SV-OOA, or LV-OOA. Detailed investigation of nitrate
behaviors revealed several factors influencing the rapid nitrate formation in summer: high ammonia emissions in the NCP region,
the gas-to-particle equilibrium of ammonium nitrate closely related to variations in temperature and relative humidity, nighttime
nitrate production through heterogeneous hydrolysis of dinitrogen pentoxide ($N_2O_5$), and regional transport from different air mass
origins. Finally, atmospheric particulate nitrate data acquired by mass spectrometric techniques from various field campaigns in
Asia, Europe, and North America uncovered a higher concentration and higher fraction of nitrate present in China. Although
measurements in Beijing during different years demonstrate a decline in the nitrate concentration in recent years, the nitrate
contribution in $PM_1$ still remains high. To effectively alleviate particulate matter pollution in summer, our results call for the urgent
need to initiate ammonia emission control measures and further reduce nitrogen oxide emissions over the NCP region.

## 32 1 Introduction

Atmospheric aerosol particles are known to significantly impact visibility (Watson, 2002) and human health (Pope et al., 2009;
Cohen et al., 2017), as well as affect climate change by directly and indirectly altering the radiative balance of Earth's atmosphere
(IPCC, 2007). The effects of aerosols are intrinsically linked to the chemical composition of particles, which are usually dominated
by organics and secondary inorganic aerosols (i.e., sulfate, nitrate, and ammonium) (Jimenez et al., 2009).
In recent years, severe haze pollution has repeatedly struck the North China Plain (NCP), and its effects on human health have
drawn increasing public attention. Correspondingly, the chemical composition, sources, and evolution processes of particulate



matter (PM) have been thoroughly investigated (Huang et al., 2014; Guo et al., 2014; Cheng et al., 2016; Li et al., 2017a), mostly
during extreme pollution episodes in winter. Unfavorable meteorological conditions, intense primary emissions from coal
combustion and biomass burning, and fast production of sulfate through heterogeneous reactions were found to be the driving
factors of heavy PM accumulation in the NCP region (Zheng et al., 2015; Li et al., 2017b; Zou et al., 2017). Although summer is
characterized by relatively better air quality compared to the serious haze pollution in winter, fine particle ($PM_{2.5}$) concentration
in the NCP region still remains high during summertime. Through one-year real-time measurements of non-refractory submicron
particles (NR-$PM_1$), Sun et al. (2015) showed that the aerosol pollution during summer was comparable to that during other seasons
in Beijing, and the hourly maximum concentration of NR-$PM_1$ during the summer reached over 300 µg m$^{-3}$. Previous studies
focusing on the seasonal variations of aerosol characteristics have noted quite different behaviors of aerosol species in winter and
summer (Hu et al., 2017). Therefore, figuring out the specific driving factors of haze evolution in summer would help establish
effective air pollution control measures.
Compared to more than 70% reduction of sulfur dioxide ($SO_2$) emissions since 2006 due to the wide application of flue-gas
desulfurization devices in power plants and the phase-out of small, high emitting power generation units (Li et al., 2017c), nitrogen
oxide ($NO_x$) emissions in China remain high and decreased by less than 20% from 2012 to 2015 (Liu et al., 2016). Therefore, the
role of nitrate formation in aerosol pollution is predicted to generally increase as a consequence of high ammonia ($NH_3$) emissions
in the NCP region. However, due to the significantly enhanced production of sulfate in extreme winter haze resulting from the
high relative humidity (RH) and large $SO_2$ emissions from coal combustion, little attention has been paid to nitrate behaviors. In
$PM_{2.5}$, aerosol nitrate mostly exists in the form of ammonium nitrate, via the neutralization of nitric acid ($HNO_3$) with $NH_3$. $HNO_3$
is overwhelmingly produced through secondary oxidation processes, $NO_2$ oxidized by OH during the day and hydrolysis of $N_2O_5$
at night, with the former being the dominant pathway (Alexander et al., 2009). The neutralization of $HNO_3$ is limited by the
availability of $NH_3$, as $NH_3$ prefers to react first with sulfuric acid ($H_2SO_4$) to form ammonium sulfate with lower volatility
(Seinfeld and Pandis, 2006). Because ammonium nitrate is semi-volatile, its formation also depends on the gas-to-particle
equilibrium, which is closely related to variations in temperature and RH. A recent review on PM chemical characterization
summarized that aerosol nitrate accounts for 16~35% of submicron particles ($PM_1$) in China (Li et al., 2017a). Some studies also
pointed out the importance of aerosol nitrate in haze formation in the NCP region (Sun et al., 2012; Yang et al., 2017). However,
detailed investigations and the possible mechanisms governing nitrate behaviors during pollution evolution are still very limited.
In this study, we present in-depth analysis of the chemical characteristics of $PM_1$ at urban sites in Beijing and Xinxiang, China
during summertime. Based on the varying aerosol composition with the increase of $PM_1$ concentration, the driving factors of haze
development were evaluated, and the significance of nitrate contribution was uncovered. In particular, we investigated the chemical
behavior of nitrate in detail and revealed the factors favoring rapid nitrate formation during summer in the NCP region.
**2 Experiments**
**2.1 Sampling site and instrumentation**
The data presented in this study were collected in Beijing from 30 June to 27 July, 2015, and in Xinxiang from 8 to 25 June, 2017.
Beijing is the capital city of China, adjacent to Tianjin municipality and Hebei province, both bearing high emissions of air
pollutants. The Beijing-Tianjin-Hebei region is regularly listed as one of the most polluted areas in China by the China National
Environmental Monitoring Centre. The field measurements in Beijing were performed on the roof of a three-floor building on the
campus of Tsinghua University (40.0 °N, 116.3 °E). The sampling site is surrounded by school and residential areas, and no major
industrial sources are located nearby. Xinxiang is a prefecture-level city in northern Henan province, characterized by considerable





industrial manufacturing. In February 2017, the Chinese Ministry of Environmental Protection issued the "Beijing-Tianjin-Hebei
and the surrounding areas air pollution prevention and control work program 2017" to combat air pollution in Northern China. The
action plan covers the municipalities of Beijing and Tianjin and 26 cities in Hebei, Shanxi, Shandong and Henan provinces, referred
to as "2+26" cities. Xinxiang is listed as one of the "2+26" cities.  The average $PM_{2.5}$ concentrations in Xinxiang in 2015 and 2016
were 94 μg m$^{-3}$ and 84 μg m$^{-3}$, respectively. Our sampling in Xinxiang was performed in the mobile laboratory of Nanjing
University, deployed in the urban district near an air quality monitoring site (35.3 °N, 113.9 °E).
An Aerodyne Aerosol Chemical Speciation Monitor (ACSM) was deployed for the chemical characterization of NR-$PM_1$, with a
time resolution of 15 minutes. Briefly, ambient aerosols were sampled into the ACSM system at a flow rate of 3 L min$^{-1}$ through
a $PM_{2.5}$ cyclone to remove coarse particles and then a silica gel diffusion dryer to keep particles dry (RH < 30%). After passing
through a 100 μm critical orifice mounted at the entrance of an aerodynamic lens, aerosol particles with a vacuum aerodynamic
diameter of ~30-1000 nm were directly transmitted into the detection chamber, where non-refractory particles were flash vaporized
at the oven temperature (~600 °C) and chemically characterized by 70 eV electron impact quadrupole mass spectrometry. Detailed
descriptions of the ACSM technique can be found in Ng et al. (2011). The mass concentration of refractory BC in $PM_1$ was recorded
by a multi-angle absorption photometer (MAAP Model 5012, Thermo Electron Corporation) on a 10-min resolution basis (Petzold
and Schönlinner, 2004; Petzold et al., 2005). The MAAP was equipped with a $PM_1$ cyclone, and a drying system was incorporated
in front of the sampling line. A suite of commercial gas analyzers (Thermo Scientific) were also deployed to monitor variations in
the gaseous species (i.e., CO, $O_3$, NO, $NO_x$, and $SO_2$).
For observations in Beijing, the total $PM_1$ mass was simultaneously measured using a PM-714 Monitor (Kimoto Electric Co., Ltd.,
Japan) based on the β-ray absorption method (Li et al., 2016). Meteorological conditions, including temperature, RH, wind speed,
and wind direction, were reported by an automatic meteorological observation instrument (Milos520, VAISALA Inc., Finland).
For measurements in Xinxiang, the online $PM_{2.5}$ mass concentration was measured using a heated Tapered Elemental Oscillating
Microbalance (TEOM series 1400a, Thermo Scientific). The temperature and RH were obtained using a Kestrel 4500 Pocket
Weather Tracker.
**2.2 ACSM data analysis**
The mass concentrations of aerosol species, including organics, sulfate, nitrate, ammonium, and chloride, can be determined from
the ion signals detected by the quadrupole mass spectrometer (Ng et al., 2011) using the standard ACSM data analysis software
(v.1.5.3.0) within Igor Pro (WaveMetrics, Inc., Oregon USA). Default relative ionization efficiency (RIE) values were assumed
for organics (1.4), nitrate (1.1), and chloride (1.3). The RIEs of ammonium and sulfate were determined to be 7.16 and 1.08,
respectively, through calibration with pure ammonium nitrate and ammonium sulfate.  To account for the incomplete detection of
aerosol particles (Ng et al., 2011), a constant collection efficiency (CE) of 0.5 was applied to the entire dataset. After all the
corrections, the mass concentration of ACSM NR-$PM_1$ plus BC was closely correlated with that of total $PM_1$ obtained by PM-714
in Beijing ($r^2$ = 0.59; Fig. S1). The slope was slightly higher than 1, which was probably caused by different measuring methods
of the different instruments and the uncertainties. For measurements in Xinxiang, the mass concentration of ACSM NR-$PM_1$ plus
BC also displayed a good correlation with $PM_{2.5}$ concentration measured by TEOM, with a slope of 0.83 ($r^2$ = 0.85; Fig. S1).
Positive matrix factorization (PMF) with the PMF2.exe algorithm (Paatero and Tapper, 1994) was performed on ACSM organics
mass spectra to explore various sources of organic aerosol (OA). Only $m/z$'s up to 120 were considered due to the higher
uncertainties of larger $m/z$'s and the interference of the naphthalene internal standard at $m/z$ 127-129. In general, signals with $m/z$ >
120 only account for a minor fraction of total signals. Therefore, this kind of treatment has little effect on the OA source
apportionment.  PMF analysis was performed with an Igor Pro-based PMF Evaluation Tool (PET) (Ulbrich et al., 2009), and the



results were evaluated following the procedures detailed in Ulbrich et al. (2009) and Zhang et al. (2011). According to the
interpretation of the mass spectra, the temporal and diurnal variations of each factor, and the correlation of OA factors with external
tracer compounds, a four-factor solution with FPEAK = 0 and a three-factor solution with FPEAK = 0 were chosen as the optimum
solutions in Beijing and Xinxiang, respectively. The total OA in Beijing was resolved into a hydrocarbon-like OA (HOA) factor,
a cooking OA (COA) factor, a semi-volatile oxygenated OA (SV-OOA) factor, and a less-volatile oxygenated OA (LV-OOA)
factor, where the former two represented primary sources, and the latter two came from secondary formation processes. In Xinxiang,
the identified OA factors included HOA, SV-OOA, and LV-OOA. Procedures for OA source apportionment are detailed in the
supplementary materials (Text S1; Tables S1-2; Figs. S2-7).
**2.3 ISORROPIA-II equilibrium calculation**
To investigate factors influencing the particulate nitrate formation, the ISORROPIA-II thermodynamic model was used to
determine the equilibrium composition of an $NH_4^+$ - $SO_4^{2-}$ - $NO_3^-$ - $Cl^-$ - $Na^+$ - $Ca^{2+}$ - $K^+$ - $Mg^{2+}$ - water inorganic aerosol (Fountoukis
and Nenes, 2007). When applying ISORROPIA-II, we assumed that the aerosol was internally mixed and composed of a single
aqueous phase, and the bulk $PM_1$ or $PM_{2.5}$ properties had no compositional dependence on particle size. The validity of the model
performance for predicting particle pH, water, and semi-volatile species has been examined by a number of studies in various
locations (Guo et al., 2015, 2016, 2017a; Hennigan et al., 2015; Bougiatioti et al., 2016; Weber et al., 2016; Liu et al., 2017). In
this study, the sensitivity analysis of $PM_1$ nitrate formation to gas-phase $NH_3$ and $PM_1$ sulfate concentrations was performed using
the ISORROPIA-II model, running in the "forward mode" for a metastable aerosol state. Input to ISORROPIA-II includes the
average RH, T, and total $NO_3^-$ ($HNO_3 + NO_3^-$) for typical summer conditions (RH = 56%, T = 300.21K) in Beijing and Xinxiang,
along with a selected sulfate concentration. Total $NH_4^+$ ($NH_3 + NH_4^+$) was left as the free variable. The variations in nitrate
partitioning ratio ($\varepsilon(NO_3^-) = NO_3^-/(HNO_3 + NO_3^-)$) were examined with varying sulfate concentrations from 0.1 to 45 μg m$^{-3}$ and
equilibrated $NH_3$ between 0.1 and 50 μg m$^{-3}$.
**2.4 Air mass trajectory analysis**
Back trajectory analysis using the Hybrid Single-Particle Lagrangian Integrated Trajectory (HYSPLIT) model (Draxler and Hess,
1998) was conducted to explore the influence of regional transport on aerosol characteristics in Beijing. The meteorological input
was adopted from the NOAA Air Resource Laboratory Archived Global Data Assimilation System (GDAS)
(ftp://arlftp.arlhq.noaa.gov/pub/archives/). The back trajectories initialized at 100 m above ground level were calculated every hour
throughout the campaign and then clustered into several groups according to their similarity in spatial distribution. In this study, a
four-cluster solution was adopted, as shown in Fig. S8.
**3 Results and discussion**
**3.1 Overview of aerosol characteristics**
Summer is usually the least polluted season of the year in the NCP region due to favorable weather conditions and lower emissions
from anthropogenic sources (Hu et al., 2017). Figures 1 and 2 show the time series of meteorological parameters, gaseous species
concentrations, and aerosol species concentrations in Beijing and Xinxiang. The weather during the two campaigns was relatively
hot (average T = 27.1 ±4.1 ℃ for Beijing and 26.9 ±4.0 ℃ for Xinxiang) and humid (average RH = 55.9 ±18.5% for Beijing
and 63.5 ±17.2% for Xinxiang), with regular variations between day and night. The average $PM_1$ (= NR-$PM_1$ + BC) concentration



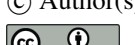

was 35.0 μg m$^{-3}$ in Beijing and 64.2 μg m$^{-3}$ in Xinxiang, with the hourly maximum reaching 114.9 μg m$^{-3}$ and 208.1 μg m$^{-3}$,
respectively. Several pollution episodes were clearly observed at the two sites, along with largely increased nitrate concentrations.
Secondary inorganic aerosol, including sulfate, nitrate, and ammonium, dominated the PM$_1$ mass with an average contribution
above 50%. The higher nitrate fraction (24% in Beijing and 26% in Xinxiang) is similar to previous observations during summer
(Sun et al., 2015; Hu et al., 2016), likely due to photochemical processes being more active than in winter. The mass fraction of
OA is lower than that measured during winter in the NCP region (Hu et al., 2016; Li et al., 2017b), in accordance with the large
reduction of primary emissions in summer. According to the source apportionment results, OA at both sites is largely composed
of secondary factors, in which 44-52% is LV-OOA and 22-23% is SV-OOA (Figs. S4-5). Primary organic aerosol accounts for
only 34% and 24% of the total OA in Beijing and Xinxiang, respectively. As there is no need for residential home-heating in
summer, which results in lower air pollutant emissions from coal combustion, chloride accounts for a smaller fraction of
approximately 1% in total PM$_1$. In addition, the higher temperature during summer drives the partitioning of semi-volatile
ammonium chloride into the gas phase, leading to lower concentrations of chloride in the particle phase.
The diurnal variations of aerosol species are similar in the measurements from Beijing and Xinxiang (Fig. S9). Organics
demonstrated two pronounced peaks at noon and in the evening. Source characterization of OA suggested that the noon peak was
primarily driven by cooking emissions, while the evening peak was a combination of various primary sources, i.e., traffic and
cooking. Relatively flat diurnal cycles were observed for sulfate, suggesting that the daytime photochemical production of sulfate
may be masked by the elevated boundary layer height after sunrise. Nitrate displayed lower concentrations in the afternoon and
higher values at night. To eliminate the effects of different dilution/mixing conditions with the development of boundary layer
height, diurnal patterns of the nitrate/sulfate ratio were analyzed to determine the role of chemical processes on nitrate behavior.
The nitrate/sulfate ratio showed the lowest value at approximately 4 pm, indicating that the evaporative loss of particulate NH$_4$NO$_3$
into gaseous NH$_3$ and HNO$_3$ overcame its photochemical production. The nitrate/sulfate ratio peaked at night, revealing the
significance of nighttime nitrate formation. During the night, nitrate production is mainly controlled by the heterogeneous
hydrolysis of N$_2$O$_5$ (Pathak et al., 2011), which is favored at high RH. A recent study conducted in urban Beijing observed high
N$_2$O$_5$ concentrations during pollution episodes and highlighted the vital role of N$_2$O$_5$ chemistry in nitrate formation (Wang et al.,

175 2017).

**3.2 Enhancement of nitrate formation during pollution episode**
To effectively mitigate aerosol pollution through policy-making, the driving factors of the PM increase need to be determined.
Figure 3 illustrates the mass contributions of various species in PM$_1$ as a function of PM$_1$ concentration in Beijing and Xinxiang.
OA dominated PM$_1$ at lower mass loadings (> 40% when PM$_1$ < 20 μg m$^{-3}$), but its contribution significantly decreased with
increased PM$_1$ concentration. The source apportionment of OA demonstrated that the large reduction in OA fraction was primarily
driven by POA, especially in Beijing. The contribution of SV-OOA and LV-OOA decreased slightly as a result of the
photochemical production. The results here are largely different from our winter study in Handan, a seriously polluted city in
Northern China, where primary OA emissions from coal combustion and biomass burning facilitated haze formation (Li et al.,
2017b). While in Beijing the contribution of sulfate increased slightly at lower PM$_1$ concentrations, the sulfate fraction generally
presented a mild decrease with elevated PM$_1$ mass at the two sites. By contrast, nitrate displayed an almost linearly enhanced
contribution with increased PM$_1$. This observation is consistent with previous summer measurements in Beijing (Sun et al., 2012)
and Nanjing (Zhang et al., 2015), China. Accordingly, the nitrate/sulfate mass ratio steadily increased as PM$_1$ went up.
Notably, the large enhancement of nitrate production mainly occurred after midnight. Figure 4 displays the scatter plots of nitrate
versus PM$_1$ as well as sulfate versus PM$_1$ for comparison, both color-coded by the time of day. Though the ratios of sulfate versus



PM$_1$ mostly increased in the afternoon, nitrate versus PM$_1$ showed steeper slopes from midnight to early morning. The correlation
of nitrate with SV-OOA and LV-OOA also indicated that the formation rate of nitrate is considerably higher than that of SV-OOA
and LV-OOA after midnight (Fig. S10). Therefore, we further checked the variations in the mass fractions of aerosol species as a
function of PM$_1$ concentration for two periods, 0:00 to 11:00 and 12:00 to 23:00. Taking Beijing as an example, both the nitrate
contribution in PM$_1$ and the nitrate/sulfate ratio were significantly enhanced for the period of 0:00 to 11:00 (Fig. S11). These results
suggest that rapid nitrate formation is mainly associated with nighttime productions, when the heterogeneous hydrolysis of N$_2$O$_5$
dominates the formation pathways along with higher RH and lower temperature. The observed high N$_2$O$_5$ concentrations in urban
Beijing further support our hypothesis (Wang et al., 2017). Because the materiality of nitrate formation to haze evolution was
observed in both Beijing and Xinxiang, we regard this as the regional generality in summer. Considering the efficient reduction in
SO$_2$ emissions in China (Zhang et al., 2012), the results here highlight the necessity of further NO$_x$ emission control for effective
air pollution reduction in Northern China.
**3.3 Factors influencing the rapid nitrate formation**
Submicron nitrate mainly exists in the form of semi-volatile ammonium nitrate and is produced by the reaction of NH$_3$ with HNO$_3$
in the atmosphere. The formation pathways of HNO$_3$ include the oxidation of NO$_2$ by OH during the day and the hydrolysis of
N$_2$O$_5$ at night. Thus, to investigate factors influencing the rapid nitrate formation in summer, the following conditions need to be
considered: (1) the abundance of ammonia in the atmosphere, (2) the influence of temperature and RH, and (3) different daytime
and nighttime formation mechanisms. Here, we explore nitrate formation processes based on Beijing measurements.
Under real atmospheric conditions, NH$_3$ tends to first react with H$_2$SO$_4$ to form (NH$_4$)$_2$SO$_4$ due to its stability (Seinfeld and Pandis,
2006). Thus, if possible, each mole of sulfate will remove 2 moles of NH$_3$ from the gas phase. NH$_4$NO$_3$ is formed when excess
NH$_3$ is available. During the sampling period, the observed molar ratios of ammonium to sulfate were mostly larger than 2 (Fig.
5), corresponding to an excess of NH$_3$. The scatter plot of the molar concentration of excess ammonium versus the molar
concentration of nitrate showed that, nitrate was completely neutralized by excess ammonium at most times. When ammonium is
in deficit, nitrate may associate with other alkaline species or be part of an acidic aerosol.
Based on the ISORROPIA-II thermodynamic model, we performed a comprehensive sensitivity analysis of nitrate formation to
the gas-phase NH$_3$ and PM$_1$ sulfate concentrations. Under typical Beijing summer conditions (T = 300.21K, RH = 56%), we
assumed that total inorganic nitrate (HNO$_3$ + NO$_3^-$) in the atmosphere was 10 μg m$^{-3}$. Total ammonia (gas + particle) and PM$_1$
sulfate concentrations were independently varied and input in the ISORROPIA-II model. The predicted equilibrium of the nitrate
partitioning ratio (ε(NO$_3^-$) = NO$_3^-$/(HNO$_3$ + NO$_3^-$)) is shown in Fig. 6. At a sulfate concentration from 0.1 to 45 μg m$^{-3}$, a 10 μg m$^{-}$
$^3$ increase of gaseous NH$_3$ generally results in an enhancement of ε(NO$_3^-$) by over 0.1 units, thus increasing the particulate nitrate
concentration. Interestingly, for ammonia-rich systems, the existence of more particulate sulfate favors the partitioning of nitrate
towards the particle phase. The formation of particulate ammonium nitrate is a reversible process with dissociation constant K$_p$:
$NH_3(g) + HNO_3(g) \leftrightarrows NH_4NO_3(s)$ (1)
K$_p$ equals the product of the partial pressures of gaseous NH$_3$ and HNO$_3$. For an ammonium sulfate-nitrate solution, K$_p$ not only
depends on temperature and RH but also on sulfate concentrations, which is usually expressed by the parameter $Y$ (Seinfeld and
Pandis, 2006):
$Y = \dfrac{[NH_4NO_3]}{[NH_4NO_3] + 3[(NH_4)_2SO_4]}$ (2)
When the concentration of ammonium sulfate increases compared to that of ammonium nitrate, the parameter Y decreases and the
equilibrium product of NH$_3$ and HNO$_3$ decreases. The additional ammonium and sulfate ions make the aqueous system favorable
for the formation of ammonium nitrate, by increasing particle liquid water content but not perturbing particle pH significantly.




Particle pH is not highly sensitive to sulfate and associated ammonium (Weber et al., 2016; Guo et al., 2017b). Therefore, more
ammonium sulfate in the aqueous solution will tend to increase the concentration of ammonium nitrate in the particle phase.
However, compared to the significant influence of gaseous NH$_3$, ε(NO$_3^-$) is weekly sensitive to the sulfate concentration, as shown
in Fig. 6. For example, when the ammonia concentration is 10 μg m$^{-3}$, a reduction of sulfate from 30 to 20 μg m$^{-3}$ has little influence
on ε(NO$_3^-$). Generally, these results suggest that a decrease in the SO$_2$ emissions may have a positive effect on nitrate reduction,
though controlling NH$_3$ emissions appears to be more effective.
The influence of temperature and RH on nitrate formation was also evaluated based on ISORROPIA-II simulations by varying
temperature and RH separately. As shown in Fig. S12, under typical Beijing summer conditions (T = 30 ℃), ε(NO$_3^-$) remains
lower than 0.1, even until RH reaches 80%. When RH > 90%, ε(NO$_3^-$) increases sharply as a function of RH. For T = 0 ℃,
representative of Beijing winter conditions, ε(NO$_3^-$) is as high as 0.7, even at low RH. Figure 7 demonstrates the variations in the
nitrate/sulfate ratio as a function of temperature and RH in Beijing. The nitrate/sulfate ratio increased with decreasing temperature
and increasing RH, which drives the nitrate partitioning towards the particle phase. This is further supported by the variations in
the equilibrium constant K$_{AN}$ of Eq. (1), which can be calculated as:
$$K_{AN} = K_{AN}(298 \text{ K})exp\left\{a\left(\frac{298}{T}-1\right)+b\left[1+ln\left(\frac{298}{T}\right)-\frac{298}{T}\right]\right\} \qquad (3)$$
where T is the ambient temperature in Kelvin, K$_{AN}$ (298) = 3.36 ×10$^{16}$ (atm$^{-2}$), a = 75.11, and b = -13.5 (Seinfeld and Pandis, 2006).
Similar to the nitrate/sulfate ratio, the diurnal profile of K$_{AN}$ peaks at night due to the lower temperature and higher RH.
As described in Sect. 3.2, the rapid nitrate formation in this study appeared to be mainly associated with its nighttime enhancement.
In addition to the effects of temperature and RH, the nighttime nitrate formation pathways may also play a role. Overnight,
particulate nitrate primarily forms via the heterogeneous hydrolysis of N$_2$O$_5$ on the wet surface of aerosol (Ravishankara, 1997).
N$_2$O$_5$ is produced by the reversible reaction between NO$_2$ and the NO$_3$ radical, where NO$_2$ reacts with O$_3$ to form the NO$_3$ radical.
Assuming N$_2$O$_5$ and the NO$_3$ radical are both in steady state considering their short lifetimes (Brown et al., 2006), the nighttime
production of N$_2$O$_5$ and HNO$_3$ is proportional to the concentration of NO$_2$ and O$_3$ ([NO$_2$][O$_3$]) (Young et al., 2016; Kim et al.,
2017). For the different PM$_1$ concentration bins, we examined the NO$_2$ and O$_3$ data at 0:00 to assess the nighttime HNO$_3$ production
rate. It can be seen that [NO$_2$][O$_3$] was obviously enhanced with an increase in the PM$_1$ mass loading (Fig. S13), implying that
nitrate formation by the N$_2$O$_5$ pathway favors the driving role of nitrate in haze evolution.
According to the Multi-resolution Emission Inventory for China (MEIC, http://www.meicmodel.org), NO$_x$ emissions localized in
Beijing are much smaller than emissions in adjacent Hebei, Shandong, and Henan provinces. In Fig. 1, episodes in Beijing,
characterized by largely enhanced nitrate concentrations, usually occurred with the change in the wind direction from north and
west to south and east, where the highly polluted Hebei, Shandong, and Henan provinces are located. When the relatively clean air
masses from north and west returned, aerosol pollution was instantly swept away. Therefore, the importance of regional transport
on haze formation in Beijing should also be considered. We examined the association of aerosol concentration and composition
with air mass origins determined through cluster analysis of HYSPLIT back trajectories. As illustrated in Fig. 8, the aerosol
characteristics are quite different for air masses from different regions. Cluster 1 mainly passed through Shanxi and Hebei provinces,
and Cluster 2 originated from Hebei, Shandong, and Henan provinces. Consistent with the high air pollutant emissions in these
areas, Cluster 1 and Cluster 2 were characterized with high PM$_1$ concentrations and high contributions of secondary aerosols. The
nitrate fraction in PM$_1$ was 24% for Cluster 1 and 26% for Cluster 2. In comparison, Cluster 3 and Cluster 4 resulted from long-
range transport from the cleaner northern areas and were correspondingly characterized by lower PM$_1$ concentrations. Organics
dominated PM$_1$ for Cluster 3 and Cluster 4, with a nitrate contribution of 14% and 16%, respectively. Figure S14 shows the cluster
distribution as a function of PM$_1$ concentration. With an increase in the PM$_1$ mass, the contribution of cleaner Cluster 3 and Cluster





4 significantly decreased. When PM$_1$ concentrations were above 20 μg m$^{-3}$, the air masses arriving in Beijing were mostly
contributed by Cluster 1 and Cluster 2, which led to rapid nitrate accumulation.

**3.4 Comparison with other regions and policy implications**

Figure 9 summarizes the chemical composition of PM$_1$ or NR-PM$_1$ (BC excluded) measured during the summer in Asia, Europe,
and North America. Three types of sampling locations were included: urban areas, urban downwind areas, and rural/remote areas.
Aerosol particles were dominated by organics (25.5-80.4%; avg = 48.1%) and secondary inorganic aerosols (18.0-73.7%; avg =
47.3%), and the nitrate contribution largely varied among different locations.  Data for the pie charts are given in Table S3.
For further comparison, we classified the datasets into three groups according to the location type and examined their difference
in nitrate mass concentrations and mass contributions. Overall, the nitrate concentrations varied from 0.04 μg m$^{-3}$ to 17.6 μg m$^{-3}$ in
summer, with contributions of 0.9% to 25.2%. Patterns in Fig. 10 demonstrate that the nitrate concentrations in mainland China
are usually much higher than those in other areas, consistent with the severe haze pollution in China. In particular, the percentage
of nitrate in aerosol particles is generally several times higher in mainland China than in other regions, except for measurements
in Riverside, CA, which were conducted near the local highway (Docherty et al., 2011). Compared to rural/remote areas, nitrate
shows higher mass concentrations and mass fractions in urban and urban downwind areas, revealing the influence of anthropogenic
emissions, i.e., traffic and power plant, on nitrate formation. In Beijing, the capital of China, field measurements among different
years show an obvious reduction in the nitrate mass concentration, especially from 2012. This coincides with the decline in satellite-
observed NO$_2$ levels in China after 2011 (Miyazaki et al., 2017) and a 21% decrease in NO$_x$ emissions from 2011 to 2015 based
on a bottom-up emission inventory (Liu et al., 2017). Detailed analysis by Liu et al. (2017) revealed that the NO$_x$ decline in China
in recent years is mainly driven by the penetration of selective catalytic reduction (SCR) in power plants and strict regulations for
vehicle emissions. The large decrease in nitrate concentration in the summer of 2008 was primarily caused by the strict emission
control measures implemented during the 2008 Olympic Games (Wang et al., 2010). However, nitrate contributions in China still
remain high over the years, especially in urban and urban downwind areas, indicating the importance of nitrate formation in haze
episodes. Overall, the higher concentration and, in particular, the higher contribution of nitrate in aerosol particles during
summertime call for the urgent need of further NO$_x$ reduction measures and NH$_3$ emission control in China.

**4 Conclusions**

Summertime field measurements were conducted in both Beijing (30 June to 27 July, 2015) and Xinxiang (8 to 25 June, 2017) in
the NCP region, using state-of-the-art online instruments to investigate the factors driving aerosol pollution. The average PM$_1$
concentration was 35.0 μg m$^{-3}$ in Beijing and 64.2 μg m$^{-3}$ in Xinxiang, with the hourly maximum reaching 114.9 μg m$^{-3}$ and 208.1
μg m$^{-3}$, respectively. Pollution episodes along with significantly enhanced nitrate concentrations were frequently observed during
the campaigns. Secondary inorganic aerosol dominated the PM$_1$ mass, with higher nitrate contributions of 24% in Beijing and 26%
in Xinxiang. The diurnal profile of nitrate presented higher concentrations at night and lower values in the afternoon. By eliminating
the influences of different dilution/mixing conditions due to boundary layer development, we found that the lower nitrate
concentrations in the afternoon were caused by the strong evaporative loss of nitrate at higher temperatures, which overcame the
daytime photochemical production of nitrate. With the development of aerosol pollution, OA showed a decreasing contribution to
total PM$_1$, despite its obvious domination at lower PM$_1$ mass loadings. The reduction in the OA mass fraction was primarily driven
by primary sources, especially in Beijing. Generally, the mass fraction of sulfate also decreased slightly as a function of PM$_1$
concentration. In contrast, nitrate presented an almost linearly enhanced contribution with the elevation of PM$_1$ mass, suggesting



the important role of nitrate formation in causing high aerosol pollution during summer. Rapid nitrate production mainly occurred
after midnight, and the formation rate was higher for nitrate than for sulfate, SV-OOA, or LV-OOA.
Comprehensive analysis of nitrate behaviors revealed that abundant ammonia emissions in the NCP region favored nitrate
production in summer. According to the ISORROPIA-II thermodynamic predictions, $\varepsilon(NO_3^-)$ is significantly increased when there
is more gas-phase ammonia in the atmosphere. Decreased $SO_2$ emissions may have co-beneficial impacts on nitrate reduction.
Lower temperature and higher RH shift the equilibrium partitioning of nitrate towards the particle phase, thus increasing the
particulate nitrate concentration. Assuming both $N_2O_5$ and $NO_3$ radicals are in steady state, $[NO_2][O_3]$ can be used as an indicator
to evaluate the contribution of nighttime $N_2O_5$ hydrolysis to nitrate formation. With the anabatic pollution levels, $[NO_2][O_3]$
obviously enhanced at night along with higher RH, suggesting the increased role of nighttime nitrate production in haze evolution.
Based on cluster analysis via the HYSPLIT model, nitrate formation was also found to depend on regional transport from different
air mass origins, in accordance with the spatial distribution of $NO_x$ emissions in the NCP region.
Finally, nitrate data acquired from this study were integrated with the literature results, including various field measurements
conducted in Asia, Europe, and North America. Nitrate is present in higher mass concentrations and mass fractions in China than
in other regions. Due to the large anthropogenic emissions in urban and urban downwind areas, the mass concentrations and mass
contributions of nitrate are much higher in these regions than in remote/rural areas. Although the nitrate mass concentrations in
Beijing have steadily decreased over the years, its contribution still remains high, emphasizing the significance of further $NO_x$
reduction and the initiation of $NH_3$ emission control in China.
Most of the previous studies conducted during wintertime reveal that secondary formation of sulfate together with primary
emissions from coal combustion and biomass burning are important driving factors of haze evolution in the NCP region. According
to this study, in Beijing and Xinxiang, rapid nitrate formation is regarded as the propulsion of aerosol pollution during summertime.
Therefore, to better balance economic development and air pollution control, different emission control measures could be
established corresponding to the specific driving forces of air pollution in different seasons. Further studies on seasonal variations
are needed to test the conclusions presented here and provide more information on haze evolution in spring and fall.
**Acknowledgements**
This work was funded by the National Natural Science Foundation of China (41571130035, 41571130032 and 41625020).

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





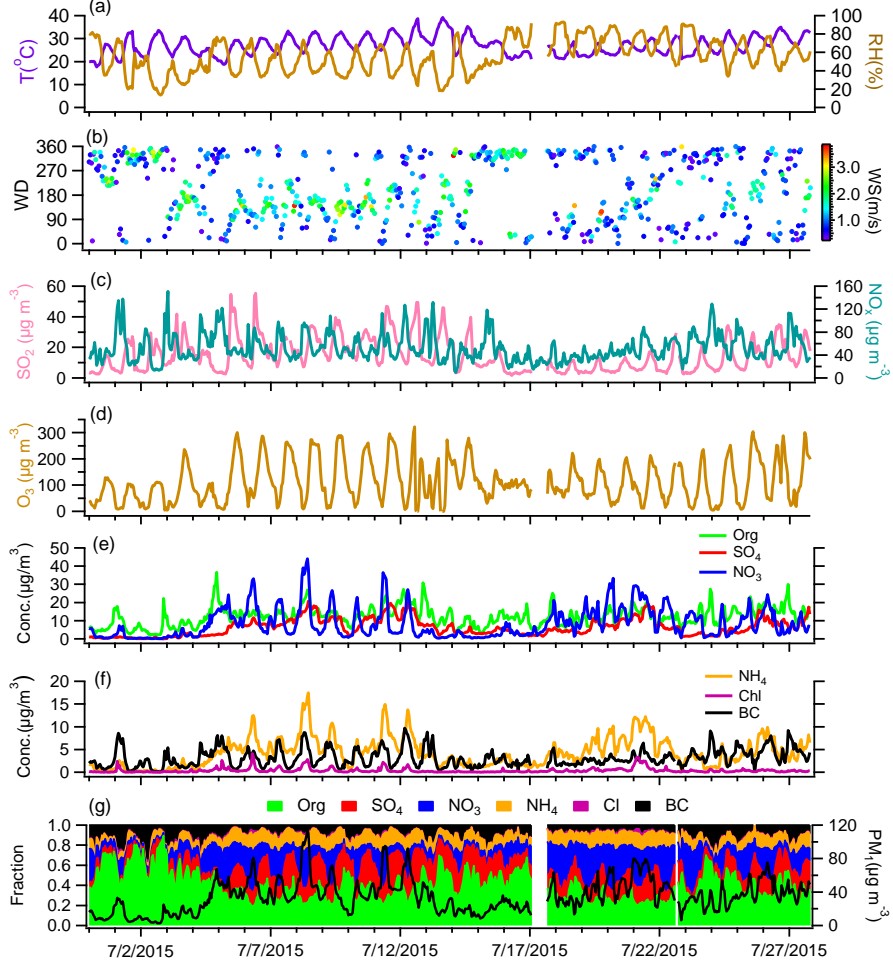

**Figure 1. Time series of meteorological parameters, gaseous species, and submicron aerosol species in Beijing.**



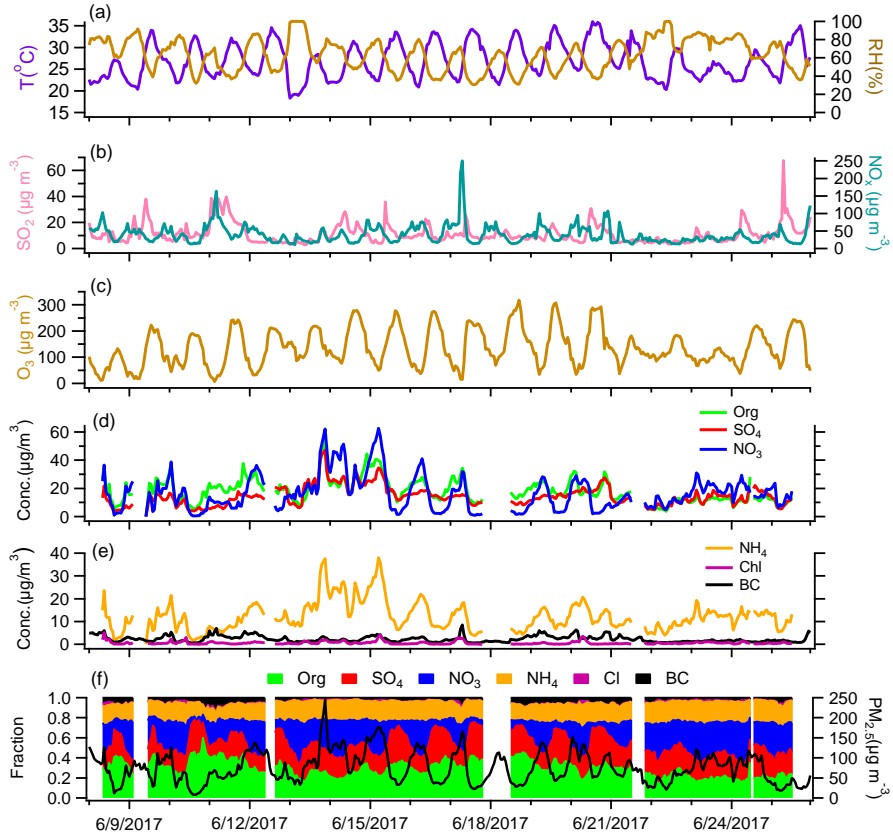


**Figure 2. Time series of meteorological parameters, gaseous species, and submicron aerosol species in Xinxiang.**





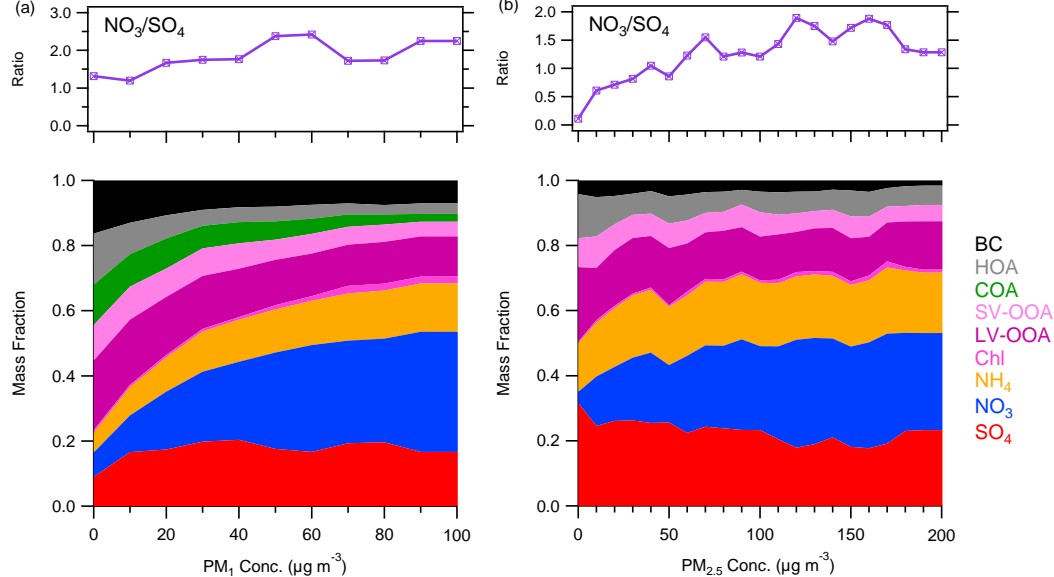


**Figure 3. Variations in the mass fraction of aerosol species and nitrate/sulfate mass ratio as a function of total PM$_1$ mass loadings in (a)**
**Beijing and (b) Xinxiang.**





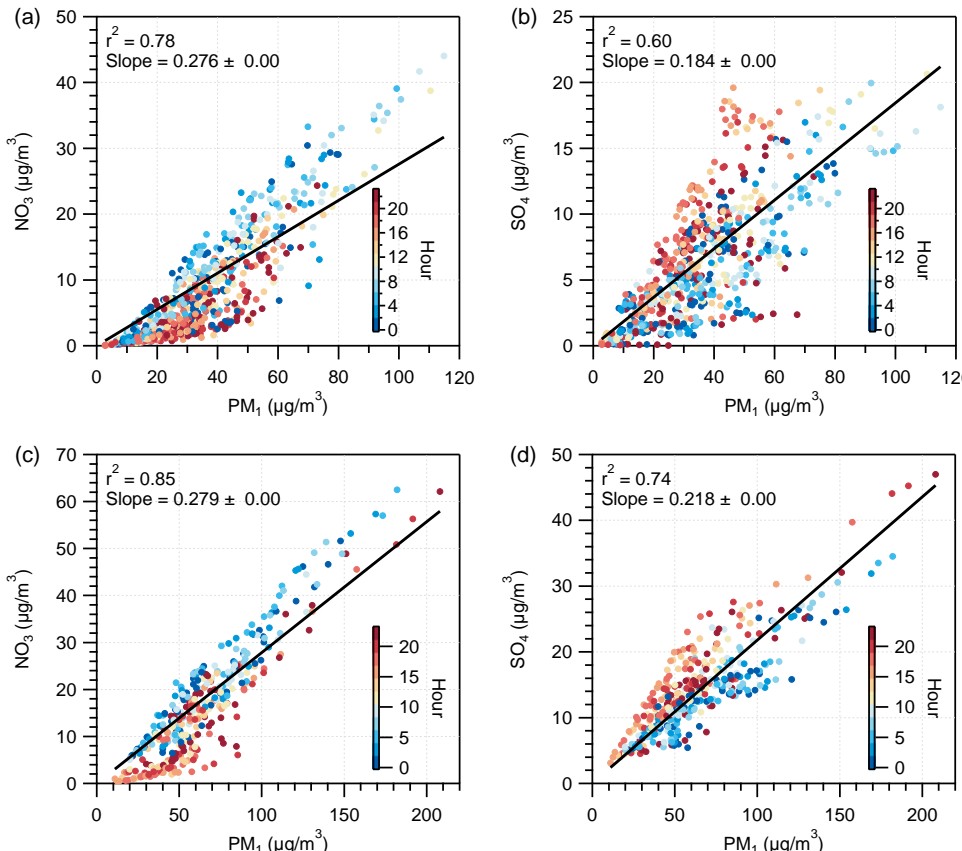


**Figure 4. Scatterplots of nitrate vs. PM$_1$ concentration and sulfate vs. PM$_1$ concentration, colored by the hour of the day, in (a-b) Beijing and (c-d) Xinxiang.**




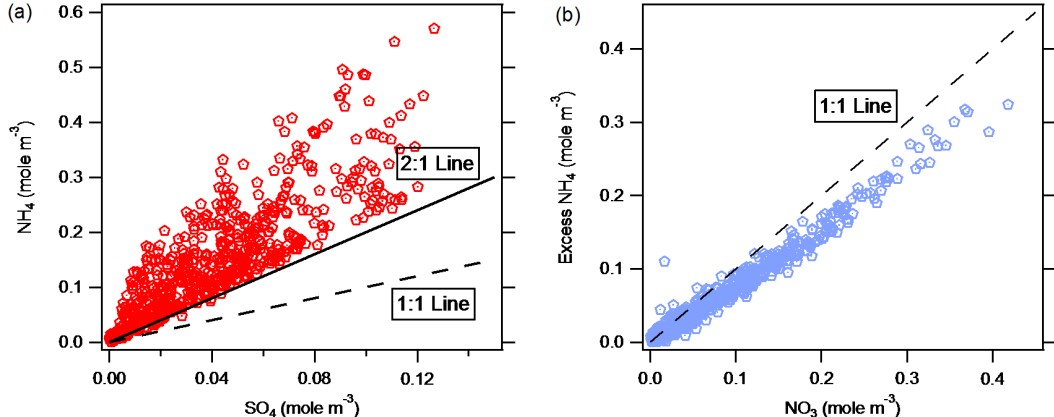


**Figure 5. Comparison of the molar concentrations of (a) ammonium and sulfate (the 2:1 reference line represents complete H₂SO₄**
**neutralization) and (b) excess ammonium and nitrate (the 1:1 reference line represents complete HNO3 neutralization).**



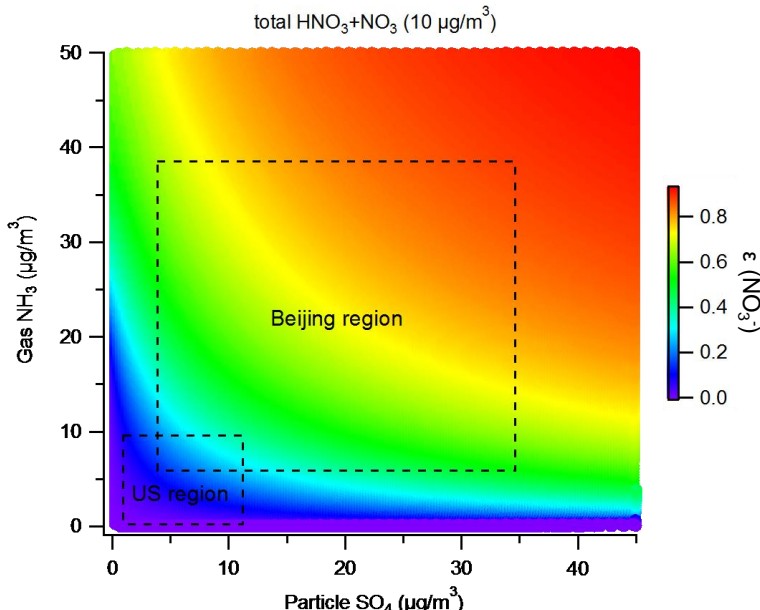


**Figure 6. Sensitivity of the nitrate partitioning ratio ($\varepsilon(NO_3^-) = NO_3^-/(HNO_3 + NO_3^-)$) to gas-phase ammonia and PM$_1$ sulfate concentrations based on thermodynamic predictions under typical Beijing and Xinxiang summertime conditions. The total nitrate concentration is assumed to be 10 µg m$^{-3}$, according to the observed PM$_1$ nitrate concentration.**






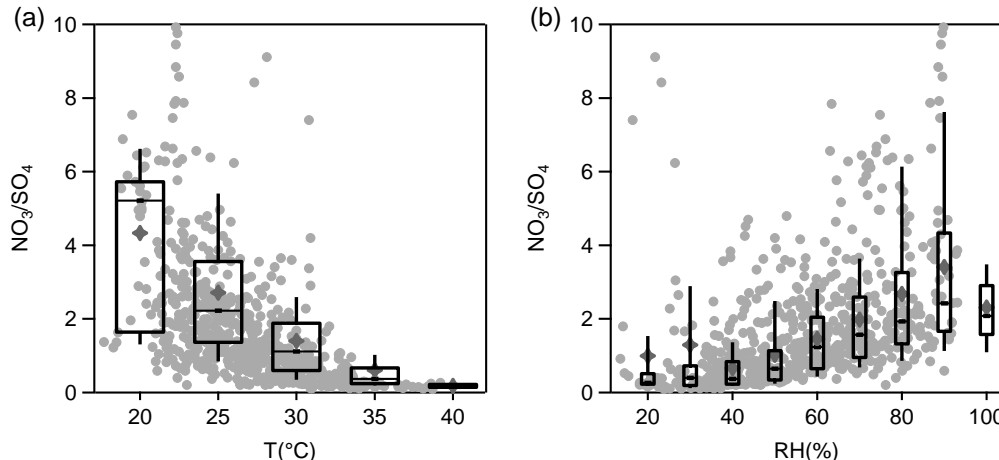


**Figure 7. Variations in the nitrate/sulfate mass ratio as a function of (a) temperature (T) and (b) relative humidity (RH). The data were**
**binned according to T and RH, and the mean (cross), median (horizontal line), 25th and 75th percentiles (lower and upper box), and 10th**
**and 90th percentiles (lower and upper whiskers) are shown for each bin.**



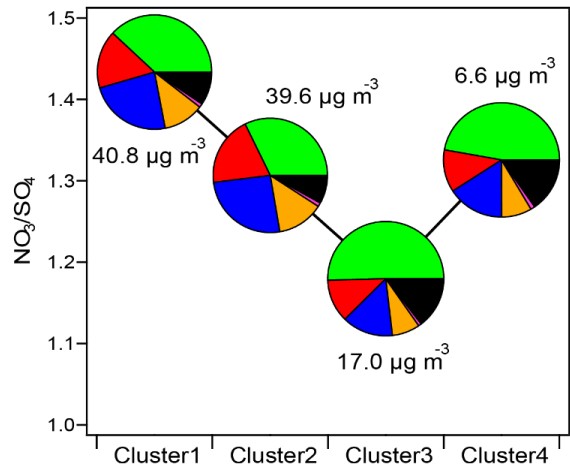


**Figure 8. Nitrate/sulfate mass ratios for each cluster. The pie charts represent the average PM$_1$ chemical composition of the different clusters. In addition, the total PM$_1$ concentrations for each cluster are also shown.**





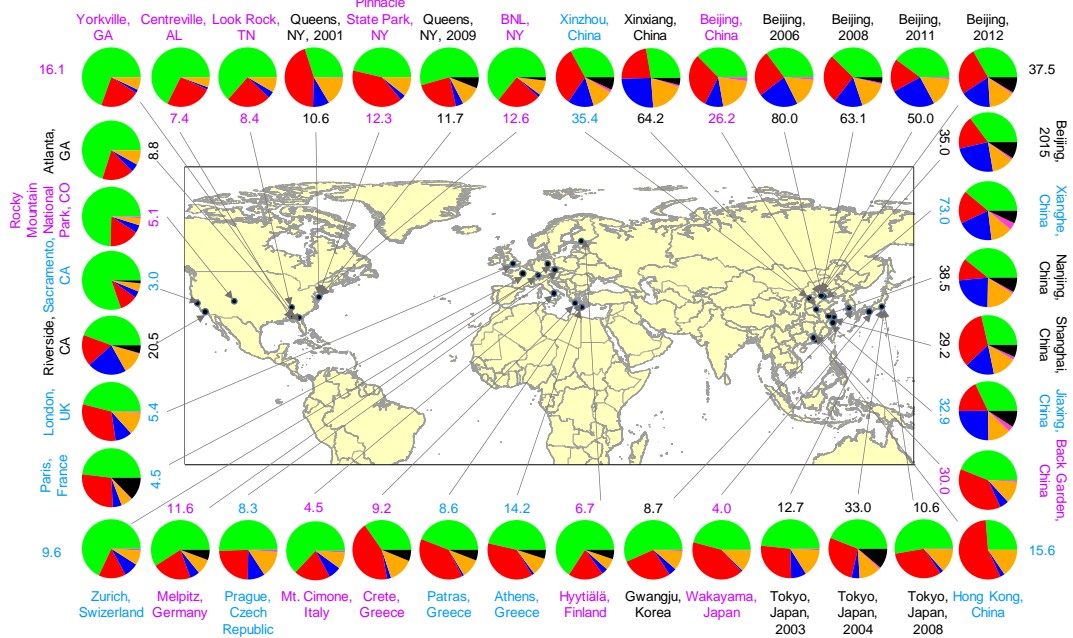


Figure 9. Summary of the submicron particle measurements in Asia, Europe, and North America (data given in Table S1 in the
supplementary materials). Colors for the study labels indicate the type of sampling location: urban areas (black), urban downwind areas
(blue), and rural/remote areas (pink). The pie charts show the average mass concentration and chemical composition of PM$_1$ or NR-PM$_1$:
organics (green), sulfate (red), nitrate (blue), ammonium (orange), chloride (purple), and BC (black).





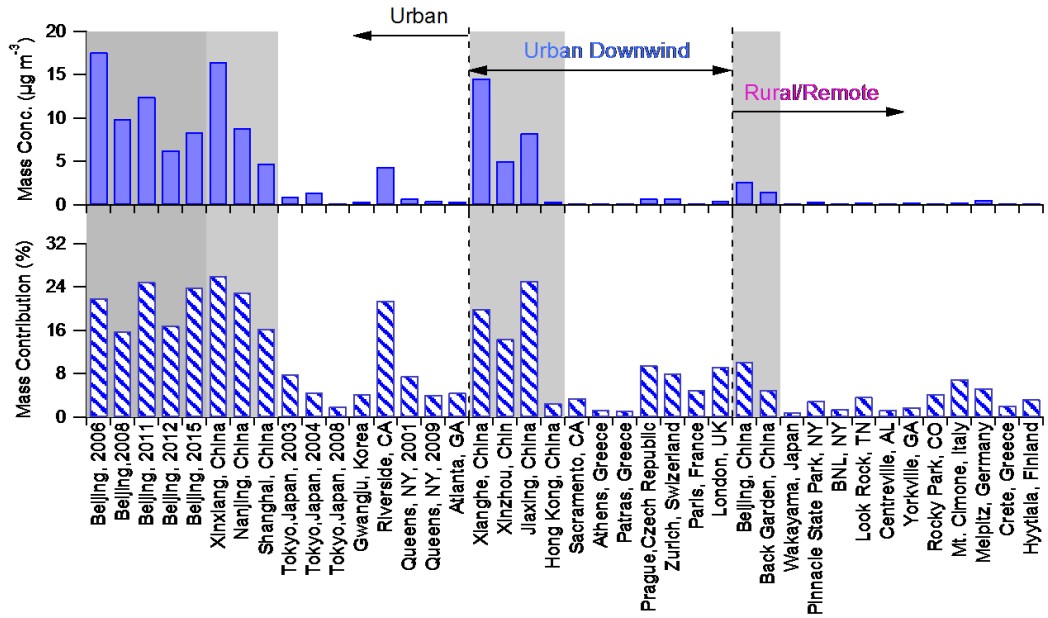

510

**Figure 10. Average mass concentrations and mass fractions of nitrate at various sampling sites for three types of locations: urban, urban downwind, and rural/remote areas. Within each category, the sites are ordered from left to right as Asia, North America, and Europe.**