# Peer review of "Nitrate-driven haze pollution during summertime over the North China 2 Plain"

_Atmospheric Chemistry and Physics, 2017_

## Referee Comment (RC1) · Anonymous Referee #2 · 29 Jan 2018

The authors investigated the sources and evolution processes of nitrate-driven haze pollution during summertime over the North China Plain. They performed field studies in two cities, and used source characterization technique and thermodynamic equilibrium model to analyze the data they achieved. The results highlight the significant role of nighttime chemistry plays in nitrate aerosol formation, and examine the influence of ammonia emissions and regional transport on nitrate yields. The work is within the scope of ACP and could be published after the following issues are addressed.

1. Page 1, line 21: Not clear what "linearly increased contribution" means.

2. Page 1, lines 24–26: The authors list four factors that influence nitrate aerosol formation, which appear equally important to readers. The text should be more precise about the main findings of this study.

[Figure]

3. Page 5, line 178: As we focus on nitrate aerosol in this study, please put nitrate at the bottom of Figure 3 to highlight the increased contribution with elevated PM1.

4. Page 5, line 188: In Figure 4, the confidence interval for the slope are all zero. Check if it is correct or not.

5. Page 6, lines 217–218: "At a sulfate concentration from 0.1 to 45 $\mu$g m$-3$, a 10 $\mu$g m$-3$ increase of gaseous NH3 generally results in an enhancement of $\varepsilon(NO3-)$ by over 0.1 units" From Figure 6, I won't say that the gaseous NH3 and $\varepsilon(NO3-)$ are linearly related. Please reorganize the sentence to be more clear and precise.

6. Page 7, lines 229–234: Sentences are not connected well. A more complete discussion of the influence of sulfate aerosol and gaseous NH3 is needed in the paper.

7. Page 7, line 260: Not clear what each colour represents in Figure 8.

8. Page 8, Section 3.4: Though this section contains links to policy implications, the discussion is very limited with only few sentences. The authors should attempt to rebalance the text in Section 3.4.

9. Page 8, Section 3.4: In Figures 9 and 10, the measurement results of this study should also be added to compare with other data. Explain what the areas of shaded regions in Figure 10 mean.

10. Page 8, Section 4: The conclusion part should be rewritten to compress text and focus on the main findings of this study. Now this section repeats the same text just as it presented above. For example, the lines 294–296 are exactly the same as lines 150–151.

---

## Referee Comment (RC2) · Anonymous Referee #1 · 14 Feb 2018

Nitrate-driven haze pollution during summertime over the North China Plain

Li et al.,

This study focus on nitrate during summertime in two cites of North China Plain. They did find nitrate concentration is higher and keep high concentration in nighttime. They tried to understand why nitrate concentration varied in daily PM1 and gave some explanation. It is interesting that they used ISORROPIA-II thermodynamic model to explain nitrate formation. Finally, they extend this study and compared with other studies in the world. The paper is suitable for the ACP. The point about nitrate is much attention for the potential readers in recent years. I would like to address one minor revision under some revisions.

[Figure]

(1) Generally, the title is not suitable. Through the whole ms, the authors only obtained data from two urban cities not rural, and background sites. "Nitrate-driven urban haze pollution during summertime over the North China Plain"

(2) Could the authors give more explanation WHY you choose the Beijing and Xinxiang? I am considering why Xinxiang is representation of urban cities. Seemly, the city is too far away from Beijing.

(3) I requested the authors shorten the section 3.1. Because the content is repeating section 3.2. For example (L153-174), nitrate concentration and production should be not given the reason here. I hope that authors carefully compared the part with section 3.2 and deleted the repeat

(4) L162 why is it ammonium chloride? There is no any citation and reason. I recommend the authors deleted the explanation. Because the chloride is too low, there is no need to give more explanation except no obvious sources in summertime.

(5) In section 3.2 L194-196 and L229-231. I might suggest the authors consider the RH and nitrate DRH here. One recent publication (Sun et al., (2018), Key role of nitrate in phase transitions of urban particles: implications of important reactive surfaces for secondary aerosol formation, Journal of Geophysical Research: Atmospheres, DOI:10.1002/2017JD027264.) They obtained nitrate-containing particles have more nitrate and lower DRH. I supposed the RH increase, these particles become mixture of liquid and solid or completedly liquid particle. The liquid-surface on particles probably promotes more nitrate-formation from the heterogeneous reactions during the nightime.

(6) L212 How to explain "nitrate may associate with other alkaline species or be part of an acidic aerosol" Maybe you need references here.

(7) It is great to compare with other studies in summer in the world. Based on Figure 9, nitrate fraction is quite higher than other places. These results might push the authors

give one conclusion to cut down the NOx emission here. I just want to mention the authors make sure all data quality is from AMS or ACMS in PM1. Maybe different technique could have different fraction here.

---

## Author Comment (AC1) · 24 Mar 2018

We thank the reviewers for their thoughtful and constructive comments. We have carefully revised the manuscript accordingly. Our point-to-point responses can be found below, with reviewer comments repeated in black and author responses in blue. Changes made to the manuscript are in quotation marks.

**Author Responses to Anonymous Referee #1**

This study focus on nitrate during summertime in two cites of North China Plain. They did find nitrate concentration is higher and keep high concentration in nighttime. They tried to understand why nitrate concentration varied in daily PM1 and gave some explanation. It is interesting that they used ISORROPIA-II thermodynamic model to explain nitrate formation. Finally, they extend this study and compared with other studies in the world. The paper is suitable for the ACP. The point about nitrate is much attention for the potential readers in recent years. I would like to address one minor revision under some revisions.

(1) Generally, the title is not suitable. Through the whole ms, the authors only obtained data from two urban cities not rural, and background sites. "Nitrate-driven urban haze pollution during summertime over the North China Plain".

Thanks for the suggestion. The title has been revised accordingly.

(2) Could the authors give more explanation WHY you choose the Beijing and Xinxiang? I am considering why Xinxiang is representation of urban cities. Seemly, the city is too far away from Beijing.

The severe air pollution in the North China Plain (NCP) has raised great concern in recent years. Henan province is an important part of the NCP region, which has experienced severe haze pollution with its rapid economic growth and urbanization. According to the China National Environmental Monitoring Center, Xinxiang was listed as one of the most polluted cities in Henan province in 2015 and 2016. The average $PM_{2.5}$ concentrations in Xinxiang in 2015 and 2016 were 94 μg m$^{-3}$ and 84 μg m$^{-3}$, respectively. To combat air pollution in the NCP region, the Chinese Ministry of Environmental Protection issued the "Beijing-Tianjin-Hebei and the surrounding areas air pollution prevention and control work program 2017" in February 2017. The action plan covers the municipalities of Beijing and Tianjin and 26 cities in Hebei, Shanxi, Shandong and Henan provinces, referred to as "2+26" cities. The 26 cities were identified according to their impacts on Beijing's air quality through regional air pollution transport. Xinxiang is listed as one of the "2+26" cities. Therefore, the field study conducted in Xinxiang would help to figure out air pollution problems in the NCP region. In addition, given the distance between Beijing and Xinxiang, the observations in Beijing and Xinxiang would help to reveal the generality and individuality of air pollution in this region. The concise introduction to Xinxiang has been given in section 2.1.

(3) I requested the authors shorten the section 3.1. Because the content is repeating section 3.2. For example (L153-174), nitrate concentration and production should be not given the reason here. I hope that authors carefully compared the part with section 3.2 and deleted the repeat.

We have carefully compared section 3.1 and 3.2, and deleted the repetitive part. Explanations on nitrate concentration and production in L153-174 has been combined into section 3.2. Detailed changes can been seen in the revised manuscript.

(4) L162 why is it ammonium chloride? There is no any citation and reason. I recommend the authors deleted the explanation. Because the chloride is too low, there is no need to give more explanation except no obvious sources in summertime.

The explanation in L162 has been deleted according to the suggestion of the reviewer.

(5) In section 3.2 L194-196 and L229-231. I might suggest the authors consider the RH and nitrate DRH here. One recent publication (Sun et al., (2018), Key role of nitrate in phase transitions of urban particles: implications of important reactive surfaces for secondary aerosol formation, Journal of Geophysical Research: Atmospheres, DOI:10.1002/2017JD027264.) They obtained nitrate-containing particles have more nitrate and lower DRH. I supposed the RH increase, these particles become mixture of liquid and solid or completely liquid particle. The liquid-surface on particles probably promotes more nitrate-formation from the heterogeneous reactions during the nighttime.

Thanks for the suggestion. The recent publication by Sun et al. (2018) revealed that ammonium nitrate content increase can reduce mutual deliquescence relative humidity (MDRH), indicating occurrence of aqueous shell at lower RH. In this study, with the enhanced nitrate formation during nighttime, nitrate/sulfate ratio increased, which would lower MDRH. Therefore, at higher RH during night, particles are supposed to be in the state of solid-aqueous or completely aqueous. The heterogeneous reactions in the liquid surface of aerosols would result in more nitrate formation. Detailed discussions have been added in the revised manuscript: "In addition, a recent study by Sun et al (2018) revealed that more ammonium nitrate content can reduce mutual deliquescence relative humidity (MDRH). With the enhanced formation of nitrate and higher RH during night, the heterogeneous reactions in the liquid surface of aerosols would result in more nitrate formation."

(6) L212 How to explain "nitrate may associate with other alkaline species or be part of an acidic aerosol" Maybe you need references here.

Aerosol nitrate can be identified mostly as ammonium nitrate but also as sodium nitrate. Unlike sulfuric acid, nitric acid has a higher vapor pressure and does not readily condense on aerosols. Therefore, the formation of aerosol nitrate requires the presence of ammonia, or other alkaline species to form salts. The related reference has been added in the revised manuscript.

(7) It is great to compare with other studies in summer in the world. Based on Figure 9, nitrate fraction is quite higher than other places. These results might push the authors give one conclusion to cut down the NOx emission here. I just want to mention the authors make sure all data quality is from AMS or ACMS in PM1. Maybe different technique could have different fraction here.

Thanks for reminding. Yes, all data in Figure 9 is from AMS or ACSM in $PM_1$. We have made it clear in the figure caption.

**Author Responses to Anonymous Referee #2**

General Comments

The authors investigated the sources and evolution processes of nitrate-driven haze pollution during summertime over the North China Plain. They performed field studies in two cities, and used source characterization technique and thermodynamic equilibrium model to analyze the data they achieved. The results highlight the significant role of nighttime chemistry plays in nitrate aerosol formation, and examine the influence of ammonia emissions and regional transport on nitrate yields. The work is within the scope of ACP and could be published after the following issues are addressed.

1. Page 1, line 21: Not clear what "linearly increased contribution" means.

The sentence has been changed to "significantly enhanced contribution" to avoid confusion.

2. Page 1, lines 24–26: The authors list four factors that influence nitrate aerosol formation, which appear equally important to readers. The text should be more precise about the main findings of this study.

Thanks for the suggestion. We have reorganized the explanation in the revised manuscript: "Based on observation measurements and thermodynamic modeling, high ammonia emissions in the NCP region favored the high nitrate production in summer. Nighttime nitrate formation through heterogeneous hydrolysis of dinitrogen pentoxide ($N_2O_5$) enhanced with the development of haze pollution. In addition, air masses from surrounding polluted areas during haze episodes also led to more nitrate production."

3. Page 5, line 178: As we focus on nitrate aerosol in this study, please put nitrate at the bottom of Figure 3 to highlight the increased contribution with elevated PM1.

Thanks for the suggestion. Figure 3 has been revised accordingly.
4. Page 5, line 188: In Figure 4, the confidence interval for the slope are all zero. Check if it is correct or not.

The slopes in Figure 4 are all zero due to the limitation of decimal places. We have corrected the problems in the revised manuscript.

5. Page 6, lines 217–218: "At a sulfate concentration from 0.1 to 45 μg m$^{-3}$, a 10 μg m$^{-3}$ increase of gaseous NH3 generally results in an enhancement of ε(NO3-) by over 0.1 units". From Figure 6, I won't say that the gaseous NH3 and ε(NO3-) are linearly related. Please reorganize the sentence to be more clear and precise.

As can be seen in Figure 6, at a sulfate concentration from 0.1 to 45 μg m$^{-3}$, the increase of gaseous $NH_3$ would lead to an increasing $\varepsilon(NO_3^-)$. The enhancement of $\varepsilon(NO_3^-)$ can be 0.1 units, 0.2 units or even higher. Because the gaseous $NH_3$ and $\varepsilon(NO_3^-)$ are not linearly related, the

sentence has been changed to make it clear: "At a sulfate concentration from 0.1 to 45 µg m$^{-3}$, a 10 µg m$^{-3}$ increase of gaseous $NH_3$ generally results in an enhancement of $\varepsilon(NO_3^-)$ by around 0.1 units or even higher, thus increasing the particulate nitrate concentration. The variations of gaseous $NH_3$ and $\varepsilon(NO_3^-)$ are not linearly related. "

6. Page 7, lines 229–234: Sentences are not connected well. A more complete discussion of the influence of sulfate aerosol and gaseous NH3 is needed in the paper.

To avoid confusion, the discussion of the influence of sulfate and gaseous $NH_3$ has been revised: "The additional ammonium and sulfate ions make the system favorable for the heterogeneous formation of ammonium nitrate, by increasing particle liquid water content but not perturbing particle pH significantly. Particle pH is not highly sensitive to sulfate and associated ammonium (Weber et al., 2016; Guo et al., 2017b). Therefore, more ammonium sulfate in the aqueous solution will tend to increase the concentration of ammonium nitrate in the particle phase. As shown in Fig. 6, at a certain concentration of gaseous $NH_3$, the increase of sulfate concentration results in a higher $\varepsilon(NO3-)$ and more particulate nitrate. Generally, these results suggest that the decreases in $SO_2$ emissions and $NH_3$ emissions are effective on nitrate reduction, indicating the importance of multi-pollutant control strategy in Northern China."

7. Page 7, line 260: Not clear what each colour represents in Figure 8.

The color legend has been added in Figure 8.

8. Page 8, Section 3.4: Though this section contains links to policy implications, the discussion is very limited with only few sentences. The authors should attempt to rebalance the text in Section 3.4.

Thanks for the suggestion. The discussion of policy implications in Section 3.4 has been extended in the revised manuscript: "Due to the installation of flue-gas desulphurization (FGD) systems, the construction of larger units and the decommissioning of small units in power plants, $SO_2$ emissions in China decreased by 45% from 2005 to 2015 (Li et al., 2017d). However, $NO_x$ emissions in China increased during the last decade. During the 11$^{th}$ Five-Year Plan (FYP), $NO_x$ emissions showed a sustained and rapid growth with the economic development and the lack of relevant emissions controls. Since 2011, the government carried out end-of-pipe abatement strategies by installing selective catalytic reduction (SCR) in power plants and releasing strict emission regulations for vehicles. Based on the bottom-up emission inventory, $NO_x$ emissions showed a decline of 21% from 2011 to 2015 (Liu et al., 2017). The changes are consistent with satellite-observed $NO_2$ levels in China (Miyazaki et al., 2017). Given the high concentration and, in particular, the high contribution of nitrate in aerosols, further $NO_x$ reduction and initiation of $NH_3$ emission controls are in urgent need in China."

9. Page 8, Section 3.4: In Figures 9 and 10, the measurement results of this study should also be added to compare with other data. Explain what the areas of shaded regions in Figure 10 mean.
The measurement results of this study has been added in Figures 9 and 10, which are labeled as "Beijing, 2015" and "Xinxiang, China". The shaded regions in Figure 10 indicate the results from China. We have made it clear in the figure caption.

10. Page 8, Section 4: The conclusion part should be rewritten to compress text and focus on the main findings of this study. Now this section repeats the same text just as it presented above. For example, the lines 294–296 are exactly the same as lines 150–151.

To focus on the main findings of this study, the conclusion has been rewritten:

[revised manuscript text omitted]